# A Review on Methods for Measurement of Free Water Surface

**DOI:** 10.3390/s23041842

**Published:** 2023-02-07

**Authors:** Gašper Rak, Marko Hočevar, Sabina Kolbl Repinc, Lovrenc Novak, Benjamin Bizjan

**Affiliations:** 1Faculty of Civil and Geodetic Engineering, University of Ljubljana, Jamova cesta 2, 1000 Ljubljana, Slovenia; 2Faculty of Mechanical Engineering, University of Ljubljana, Aškerčeva cesta 6, 1000 Ljubljana, Slovenia; 3The National Institute of Chemistry, Hajdrihova ulica 19, 1000 Ljubljana, Slovenia

**Keywords:** free water surface, measuring methods, laser scanning, high-speed imaging, two-phase flow, turbulent flow

## Abstract

Turbulent free-surface flows are encountered in several engineering applications and are typically characterized by the entrainment of air bubbles due to intense mixing and surface deformation. The resulting complex multiphase structure of the air–water interface presents a challenge in precise and reliable measurements of the free-water-surface topography. Conventional methods by manometers, wave probes, point gauges or electromagnetic/ultrasonic devices are proven and reliable, but also time-consuming, with limited accuracy and are mostly intrusive. Accurate spatial and temporal measurements of complex three-dimensional free-surface flows in natural and man-made hydraulic structures are only viable by high-resolution non-contact methods, namely, LIDAR-based laser scanning, photogrammetric reconstruction from cameras with overlapping field of view, or laser triangulation that combines laser ranging with high-speed imaging data. In the absence of seeding particles and optical calibration targets, sufficient flow aeration is essential for the operation of both laser- and photogrammetry-based methods, with local aeration properties significantly affecting the measurement uncertainty of laser-based methods.

## 1. Introduction

Precise and reliable measurements of complex hydraulic phenomena including high-speed flow and particularly aerated flow features are essential for obtaining new information and knowledge about water flow structures and accompanying processes [1]. Comprehensive knowledge of these phenomena combined with the capability for their accurate modelling is required to avoid under-dimensioned or otherwise poorly constructed hydraulic structures such as high-speed outlets, spillways of HPP, sediment bypass tunnels, desilting and fish migration facilities, drop shafts, etc. [2]. The knowledge of hydraulic phenomena is well advanced mainly in subcritical flows but is still limited in supercritical flows and transitions to two- or multi-phase flows [2]. Such flows are present in a wide range of applications in civil, chemical, environmental, mechanical, mining and nuclear engineering.

Most free-surface flows in hydraulic structures are turbulent and characterized by a varying degree of air bubble entrainment due to surface deformation and high shear stresses that exceed surface tensions resisting interfacial breakup [3]. Free-surface flows with high Reynolds and Froude numbers (approximately Re > 10^4^ and Fr > 3, respectively) are highly complex, non-homogeneous and non-stationary [4,5]. The surface is undulating, and velocity variations change the height according to Bernoulli equation. Due to shear forces often prevailing over surface tension, surface break-up is frequent, leading to the formation of both water-entrained bubbles and droplets flying above the surface [6]. A high Reynolds number of the flow causes bubbles and droplets to form a very prominent non-spherical shape and high velocity [7]. Another interesting feature of flows at high Reynolds and Froude numbers is their ability to form large standing and fluctuating quasi periodic waves at confluences [8]. Standing wave heights at the confluences are often several times higher than both flows in front of the confluence. The surface of standing waves features similar properties as the aforementioned aerated flows with a heavily undulating and uneven surface, many entrapped bubbles and flying droplets. Standing waves at the confluence may also form a concave surface, which is difficult to capture using available measurement methods, often preventing the unambiguous determination of the free-surface position and height.

Besides the evident necessity of correctly dimensioning hydraulic structures to prevent overflowing or excessive recirculation when complex turbulent and highly aerated flows are expected, another important design consideration is the impact of turbulent water mixing, hydraulic jump and consequent water aeration, which affect the ecological conditions in surface water bodies [9]. Oxygen concentration in the water is very important when determining surface water quality. Turbulent mixing helps to minimize the oxygen deficit and encourages positive changes in the microbiological metabolism of the flow [10].

The conventional approach to hydraulic structure design has been to rely upon fundamental hydrodynamics and theoretical studies, combined with a range of historically proven designs [3]. Often, this has included construction of scaled-down models with manageable dimensions so that classic flow measurement methods (e.g., manometers, wave probes and point gages) could be utilized to determine the water flow behavior. Although adequate for simpler hydraulic structures that had been constructed for many centuries, such a simplistic design strategy often cannot meet the requirements of demanding supercritical flow designs as it involves significant trial-and-error which is costly, time-consuming and potentially dangerous under unforeseen extreme operating conditions (e.g., flooding, erosion, etc.). The greatest design challenges are experienced when high-speed and/or highly aerated flows are involved [2].

To reduce the cost and risk associated with the design and operation of hydraulic structures, the accurate modeling of water flow and related phenomena is of great importance. Modern design methods include both experimental measurements on scaled-down hydraulic models, as well as numerical simulations by computation fluid dynamics (CFD). The capability of modern CFD simulations to simultaneously calculate many flow-related quantities (velocity, pressure, volume fraction…) in high resolution is certainly attractive, especially when simulation results are presented in a visually appealing manner. Nevertheless, the accuracy and reliability of numerical simulations are still questionable, since simulation results can be very sensitive to the wide range of computational parameters that have to be set by the user [6]. Additionally, each CFD model requires calibration and verification by experimental measurement methods, and often the amount of quality measurement data is insufficient to do so. For this reason, the development of hydraulic measurement techniques with high spatial and temporal resolution is a priority to improve the surface measurements and turbulent surface flow modelling in conjunction with CFD simulations. Without doubt, a great deal of insight into the operation of hydraulic structures has been gained over decades of using flow measurement methods such as point gauges and manometers. On the other hand, immense advances in the last two decades in laser- and photogrammetric-based methods for terrestrial scanning have also led to the limited adaptation of these methods for the measurement of free-surface flows [11].

This review paper aims to review both conventional and modern methods with an understanding that no single method can be universally applied to all types of free-surface flows. Limitations in instrumentation as well as the complexity of flow structure and dynamics (internal dynamic pressure gradients, shear forces, air entrainment and entrapment, etc.) often require careful consideration when attempting to determine the optimal measurement method for the flow problem at hand. As discussed above, flows with high Reynolds (above 10,000) and Froude (above 3) numbers with free surface are highly complex, non-homogeneous and non-stationary. In addition, they can be characterized by undulating surface break-up, high local velocities, the non-spherical shape of bubbles and droplets, the formation of standing quasi-periodic waves with exceptional relative heights, etc. The review includes conventional and more recent advanced optical methods to address the properties of the turbulent aerated flow with free surface. The selection of available measurement methods is large and, for instance, optical methods such as LIDAR and photogrammetry may be both used in their aerial and terrestrial forms, both featuring their own characteristics and requirements. To make the review short and concise, we will only focus on the terrestrial methods. Additionally, we will try to provide a critical assessment of the advantages and disadvantages of each method.

## 2. Methods for Measuring Free Water Surface

### 2.1. General Classification

Numerous different methods exist for the purpose of measuring the free water surface. These methods can be classified with respect to their intrusiveness (contact vs. non-contact), spatial resolution (point measurements vs. 2D/3D spatial measurements) and temporal resolution (low vs. high sampling rate). Method classification and comparison are provided in Table 1. 

Static water levels with little or no aeration present are mostly measured by well-established flow measurement devices such as U-manometers [12], point gauges [13,14], wave probes [15,16] and ultrasonic sensors [17,18]. On the other hand, flows with an aerated air–water surface are most frequently analyzed by the means of ultrasonic sensors and high-speed video cameras in a side-view perspective, while laser-based technologies have also received increased research attention in recent years [19]. Free-water-surface methods will be reviewed in detail in the following chapters.

### 2.2. Manometers, Wave Probes, Point Gauges

Steady water flows with a negligible presence of aeration allow for the use of rather inexpensive and well-established measurement methods such as manometers, wave probes and point gauges (Figure 1). Manometers are among the oldest pressure measurement instruments and operate by indicating differential pressure proportionally to the liquid column weight due to the difference in liquid pressure on both ends of connecting tubes. Most commonly, manometers are designed in the form of a U-tube where measured pressure difference is proportional to the difference in liquid height between the left and right vertical tube section. Another well-known design is a bulb manometer where the liquid in a tube is open to the measured medium on one side while it seals the reference gas contained within a bulb on another side. Daood et al. [20] performed a comparative study of a concentric tube bulb manometer and a U-tube manometer. Note that an operating principle such as a manometer is also implemented by piston gauges (also known as pressure balances), albeit with a different mechanism of producing the pressure difference readout. The measurement uncertainty of U-tube manometers and pressure balances was investigated by Stuart [21]. Although manometers are not well suited for the measurement of highly aerated flows, certain flow problems can benefit from the measurement of the air phase. Teng and Yang [12] employed high-sensitivity differential manometers in spillway design for the purpose of measuring the air pressure in the air cavities below the water jets in flip-bucket aerators. Typical measurement uncertainty is around ±1 mm [22].

The two main operational challenges of manometer depth measurement are errors incurred by the effect of flow velocity and presence of bubbles or impurities. When velocity of the fluid *v* above the manometer probe is higher than zero, according to Bernoulli equation (Δ*p* = *ρgh*+ *ρv*^2^/2) the measured water surface height decreases (Figure 1, left). Unfortunately for the operator, the velocity is not always known so that its effect could be compensated in calculations. In the cases of interest for this review, velocity spatially may greatly vary over the observation region. Thus, surface height variations are usually unknown. Complicating situation even further, velocity may vary with height above the surface (*h* = *v*^2^/2*g*). Additionally, any bubbles trapped within the fluid decrease the effective density $\bar{\rho }$ of the fluid, and hence underestimate the height $h=\Delta p/\bar{\rho }g$ of the surface level. On the other hand, sediments within the fluid by their higher density underestimate the measured height of the surface level. Due to slow response, the manometer is unsuitable for measuring a fluctuating water surface.

Wave probes are intrusive resistance-type devices that measure water depth as proportional to the resistance between two parallel immersed electrodes (usually stainless steel) and can achieve measurement uncertainty of less than ±1 mm in flows with smooth water surfaces [22]. To prevent electrolysis (problematic due to the forming of gas and electrode corrosion), an AC drive from a low impedance current amplifier is typically used. Besides the liquid depth, incident and reflected pressure waves (gravity waves, etc.) can also be detected when multiple wave probes are installed at a given distance from each other [15,16] The issues with gas presence and nonzero velocity can be partly addressed by advanced probe designs that are also able to detect phase or measure velocity. A special wave probe variant is a phase detection probe used in multiphase (e.g., aerated) flows to detect whether there is water or air in the measurement zone [23,24]. Dual-tip phase-detection probes are also capable of measuring flow velocity in aerated water flows [25,26]. Nevertheless, it is optimal that wave probes are installed in regions with low velocity whenever possible since higher velocities lead to the formation of stagnation points at the probe (hence, overestimated depth due to local pressure increase), downstream wake (underestimated depth), vortex shedding, probe-wetting issues, variations in conductivity due to the presence of bubbles, etc. These problems appear because of the presence of velocity in horizontal directions and fluctuations in vertical directions.

The water surface level can be determined by point gauges whereby the surface height is established visually from the instrument scale. Pfister and Gisonni [13] performed point gauge water level measurements to analyze head losses in free-surface flows through sewer junctions. Based on measurement results, models for energy loss prediction were formed as a function of upstream and lateral conduit diameters and flow conditions. The point gauge measurement uncertainty is of the order of magnitude of ±1 mm, with an accuracy of up to 0.1 mm being achievable when water surface is perfectly still [22]. If the gauge tip touches the surface, measurement uncertainty is increased as the surface tension causes droplets to adhere to the tip. The main advantage of the point gauge method lies in the simplicity and accuracy of the method, with measurement uncertainty also of the order of magnitude of ±1 mm, as in the case of manometers and wave probes, meaning that all these methods can be used as a reference for other, possibly more advanced measurement methods [22]. Nevertheless, all three methods have the shortcoming of being rather time-consuming and do not allow for the automated continuous measurement of the water surface profile. The methods also perform poorly in the measurement of water depths when the flow is aerated or exhibits strong height fluctuations [22].

### 2.3. Electromagnetic and Ultrasonic Methods

Electromagnetic and ultrasonic measurement techniques allow for the nonintrusive measurement of water flow topography and velocity conditions. 

The electromagnetic field is known to affect water, especially when laden with particles or sludges [27], but the opposing effect (movement of water causing changes in detected electromagnetic field) also applies. The electromagnetic sensors are based on Faraday’s law of electromagnetic induction measuring the velocity of a conductive liquid moving through a magnetic field [28]. Electromagnetic sensors have been used for many decades to measure turbulent velocities and oscillatory flow in marine and river environments. Their main advantages are robustness, resistance to particles or air contamination and a good frequency response in the range below 20 Hz [29], although their relatively large size prohibits small-scale measurements. In contrast, ultrasonic sensors (also known as acoustic displacement meters) usually operate on the principle of frequency change of emitted sound as it propagates along a flow path. Flow velocity can be calculated from the sound propagation velocity (approximately 1500 m/s for unaerated water). Ultrasonic sensors have a good frequency response of up to 30 Hz, which is somewhat higher than electromagnetic sensors but are more sensitive to air and particle contamination and thus more suitable for a controlled laboratory environment [28]. However, due to the single point measurement principle and size of the signal spot, their spatial resolution is limited [22] and a blind zone [18] prevents measurements near the sensors due to the time needed for a transducer to switch from the sending to receiving mode of operation. Additionally, unlike laser-ranging devices, ultrasonic distance meters are usually fixed as the speed of sound is much lower than the speed of light, thus preventing the high-frequency measurements of free water surface. Consequently, several fixed-position sensors are often employed for distance measurements [30] and can serve as calibration for high-resolution methods such as LIDAR or high-speed imaging. 

Capabilities and limitations of ultrasonic sensors were investigated in [18] for several types of turbulent flows by synchronization with a high-speed camera. Ultrasonic sensing was found to adequately reproduce free-surface dynamics in predominantly two-dimensional flows with low-frequency depth oscillations (i.e., below the sensor‘s Nyquist frequency). However, ultrasonic sensors tend to perform poorly in highly turbulent and aerated three-dimensional flows (e.g., in aerated spillways). The broad spectrum of time scales in such processes cannot be adequately resolved from heavily aliased ultrasonic sensor signals, and other instruments with higher sample rates are preferable [18]. In the cases of an inhomogeneous mixture and high three-dimensional flows such as the aerated stepped chute flow, ultrasonic sensors fail to reproduce the three-dimensional flow characteristics, merely enabling the reproduction of average characteristics over the entire sampling surface. The authors [18] also discussed the application of ultrasonic sensors to characterize entrapped air in high-velocity flows. Although high-speed imaging by itself is a very important free-surface flow measurement method and will as such be discussed in a separate chapter, it is worth noting that several authors have complemented it with ultrasonic measurements to gain additional insight into the mechanism of turbulent multiphase flows. Papers [17,30] employed a combination of both methods to detect the air–water surface roughness in self-aerated chute flows through stepped spillways, while study [31] used a similar setup to evaluate the free-surface characteristics of hydraulic jumps. As observed in [30], the optimal arrangement of acoustic displacement meters is normal to the free surface of interest, while measurement uncertainty and percentage of rejected data increases significantly if the detection zone axis is not perpendicular to the surface, or if the surface has a low curvature radius in the detection zone. 

### 2.4. Light Detection and Ranging (LIDAR)

The light detection and ranging (LIDAR) method is currently one of the most widely used and promising remote sensing technologies. Most LIDAR devices operate by emitting a laser beam (typically pulsed) towards the surface of interest and measuring the time of flight of reflected light to determine distance to the surface [32]. Simpler LIDAR devices have a fixed position of beam emitter and receiver, allowing for point measurements, whereas more advanced devices are capable of periodically scanning the surface in one or two dimensions by utilizing rotating mirrors, linear motors or other beam manipulating elements. In the case of scanning LIDAR or similar devices, the measuring frequency and laser beam scanning speed can be on the order of 100 kHz and several hundred profiles per second, respectively, meaning that water surface topography and dynamics can be measured with a reasonably high resolution only exceeded by high-speed imaging [11]. Because of its speed, accuracy, and efficiency, airborne and terrestrial LIDAR scanning is increasingly replacing traditional geodetic methods of measurement [33,34]; it facilitates the capture of data required for archaeological research [35,36] and detailed reconstruction of buildings [37,38], and with the classification of raw point clouds with enhanced algorithms, the data can be used in forestry [39,40,41], geomorphology [42], laser bathymetry [43], etc. LIDAR-acquired topography data are also widely used in hydroengineering, particularly for preparing geometry in physical and numerical modeling [44,45,46,47], with a notion that water bodies usually greatly affect the performance and accuracy of measurements. In these applications, water represents a layer that must be penetrated by the laser beam to obtain the topography of an underlying solid surface.

If, on the other hand, the water surface shape itself is of interest, the measurement methodology must be adjusted to reduce reflections from other surfaces (e.g., channel bed), as well as immersed air bubbles and solid particles unless very near water surface. Reflections from water surface can be of either specular or quasi-diffuse type [22]. Reflections from water surface are remarkably different than reflections from solid objects (all diffuse-reflecting surfaces such as walls, plants, soil, people). Individual reflections from water surface, bubbles or droplets are all specular. Figure 2 shows specular reflections from droplets or bubbles of near circular and distorted shape, and in flows with high Reynolds numbers, bubbles with distorted (non-circular) shape prevail. Apart from the Reynolds number, liquid viscosity and surface tension are also known to affect the bubble formation process, particularly bubble diameter, shape and rising velocity [48].

Speed of light is many times higher than velocity of the airflow, hence, consecutive specular reflections from bubbles and droplets occur while the flow is perfectly stationary, as shown in Figure 3 (middle and right). Some of the reflections may find a way to the lidar beam reception optics. Therefore, we argue that a series of reflections from individual bubbles may be regarded as a quasi-diffuse reflection. This is because LIDAR beam is of the same size as the bubbles and droplets or larger. On the contrary, reflections from common solid objects are most often diffuse. 

Different types of reflections are compared in Figure 3. For the case of specular reflection (Figure 3, left), the incident laser beam is reflected from the water surface in a single outgoing direction and can only be detected by the receiver for a narrow range of reflection angles [22]. This is a typical situation for still water, where light may be reflected from the surface, or penetrate through liquid, instead reflecting from the bottom, bubbles, or immersed particles [32]. In the event of quasi-diffuse reflection, the beam is scattered in a wide-angle pattern upon reflection from the surface, and often returns to the device receiver. Provided that the intensity of reflected light is sufficient, the distance from the surface can be measured. In aerated flows, quasi-diffuse reflection is predominant due to the presence of air bubbles close to the water surface [22].

The concept of laser ranging of the water surface to measure its topography is quite recent, as it was first applied by Blenkinsopp’s group circa 2010. They used LIDAR to measure the time-varying free-surface profile across the swash zone [49]. The water surface can foam in the swash zone’s area, which increases the probability of diffuse reflections, and hence, successful measurements with LIDAR. The results show good agreement with ultrasonic sensor measurements. Another group [50] applied the method for laboratory profile measurements of the time-varying free water surface of propagating waves. Both groups have performed measurements in wave flumes and have used water mixed with a particulate matter to improve reflections. High-frequency measurements of wave transformation and characteristics using a LIDAR scanner were also performed in a lab environment [51] and in coastal maritime regions [52,53,54]. LIDAR technology was also applied for analysis of aerated hydraulic jumps [55,56,57], where free-surface features were obtained with high spatial and temporal resolution in fully aerated regions. This provided new insights into the interactions between the aerated hydraulic jump toe and the free-surface features, including the large-scale free-surface motion and the relationship between the hydraulic jump toe oscillation and the fluctuations of the free water surface along the turbulent aerated region. Using advanced post-processing of the raw LIDAR point cloud based on autocorrelation and cross-correlation analyses, they were able to estimate the time and length scales of the free surface in fully aerated hydraulic jumps. The results of the research clearly demonstrate the importance of continuous LIDAR measurements with high spatial and temporal resolution. Anyway, they suggest simultaneous measurements combining different measurement techniques as well as filtering methods to further investigate the length scales of free water surface.

Laser scanning also provided free-surface features in fully aerated flows over a stepped spillway and down a laboratory still basin at high spatial and temporal resolutions [19,46]. Both studies highlight the applicability and capability of LIDAR to provide information in the field of physical modelling and to provide data with improved spatial and temporal resolution, enabling design improvement through detailed characterization of the free surface of complex air–water flow motion, as well as other advantages over traditional point measurements in situ. Results from Kramer’s think tank research showed good agreement with characteristic air–water flow heights and internal flow parameters compared to previous acoustic Doppler current profiling. The application of various correlation analyses enabled them to obtain integral time scales of the two-phase flow along the gradually varied flow region. They note that the relatively low sampling rate of the LIDAR instrument used was the main limitation. This issue has been largely resolved by the advent of commercial instruments with a much higher sampling frequency and angular resolution that have been successfully used by other groups for free-water surfaces (e.g., [32]).

Rak et al. [58] applied laser scanning to measure turbulent, foamed free-water-surface flow at the hydraulic structure in the tailwater channel from the second power unit of the Doblar I hydroelectric power plant on the river Soča (Slovenia). Time-varying free-surface profile of the water surface and the range of water surface fluctuation was measured with high spatial and temporal resolution. Li et al. [59] used LIDAR to acquire mean free-water-surface profiles of the aerated creek flows. The authors measured free water surface in aerated conditions and bathymetry during low discharges and consequently non-aerated conditions to generate 3D channel bed and free-water-surface maps. An overview of LIDAR-based free-surface measurement studies is provided in Table 2.

Another and likely the most challenging measurement area where the LIDAR method was implemented is the supercritical confluence flow. Rak et al. [22,32,60,61] investigated the applicability and capability of this method on a 90° open channel junction that allowed to generate highly complex, three-dimensional and aerated standing waves. A scanning LIDAR moving on rails was utilized to measure the average water surface profiles and fluctuations, and a 3D average surface was reconstructed. Reference measurements were provided visually by a high-speed camera and a thin slice ruler inserted in the water within the field of view. Although LIDAR measurement results were in good agreement with visual measurements, verification was only approximate since no practical reference method exists that can measure the complex standing wave topography with comparable resolution to LIDAR and in a reasonably short time.

To further evaluate the performance of LIDAR for measuring aerated flows, Rak et al. [22] performed controlled condition measurements in a glass tank where otherwise still water was aerated with bubble injection, and thin soap foam film was added under some conditions. Results showed that in non-aerated water effective measurements are limited solely to points where the laser beam is reflected directly back to the scanner at an angle of 0° (difficult to provide and control in experiments). Both bubble injection and soap foam additionally increased the number of reflections. The incipient laser beam was diffusely reflected off the bubbles on the surface as well as off the bubbles below the water surface, while bubble injection also increased the range of angles at which the laser scanner received echoes. A low degree of aeration was shown to cause significant reflections from submerged bubbles, leading to an overestimation of distance to the water surface. Additionally, as noted by Pope and Fry [62], lasers with a shorter wavelength penetrate the water slightly better so measurements can have significant noise. In confluence-flow experiments [22], a comparison between acquired LIDAR-acquired surface profiles and reference visual measurements showed that the proper course of transverse water-surface can be obtained despite up to ±20% fluctuations in the average water depth. With corrections based on the remission values of received echoes, the agreement between water levels was very good (a 10 mm order-of-magnitude deviation for water depths of up to 300 mm).

Based on studies referenced in this section, it can be concluded that LIDAR technology can, under appropriate conditions, be used for measuring water levels even without adding particulate matter to improve the number of signal returns, provided the flow is sufficiently aerated (e.g., hydraulic jumps, flow over stepped spillways, confluence flows, etc.) and acquired ranging data are filtered to account for the specifics of beam reflection in the given application to minimize the distance measurement error.

### 2.5. Laser Triangulation

Over the past two decades, flow imaging devices and image processing algorithms have made tremendous progress, facilitating the development of image-based technologies for measuring the free surface of flows in open channels. This progress has also been applied to the measurement of open-surface turbulent flows. The two main techniques used for this purpose are triangulation and photogrammetry [63].

The basic operating principle of the laser triangulation method is to emit one or more laser beams onto the fluid surface to be observed and then to record the positions of impact and reflection on said surface with a camera [64]. Although laser triangulation appears to be similar to the LIDAR method, it does not operate on the time-of-flight principle but uses trigonometric relationships to determine the position of the surface from which the laser reflections originate [11]. In addition, the reflected light is captured by a camera rather than a single-point light receiver, allowing for many reflections to be captured in a single image.

While the laser triangulation technique is commonly used for scanning solid surfaces, work on triangulating water surfaces in highly turbulent and aerated flows is relatively limited. Mulsow et al. [65,66] used a triangulation method in which the reflected laser line is captured by a camera to determine the elevation profile of a non-aerated flow, and applied the laser triangulation method to free-surface measurements of high-turbulence and aerated confluence flows in a 90° channel bifurcation with high Reynolds and Froude numbers (Figure 4). A comparison with measurements from LIDAR, made simultaneously with the same laser instrument, showed that the performance of both methods was comparable in measuring the height of turbulent open surfaces and in estimating the average height of the turbulent open surface.

Due to multiple reflections from the bubble and droplet surfaces of the two-phase flow, each method used for calibration or validation can give significantly different results, so comparisons can be difficult. An example of validation using the camera triangulation reference method in Figure 4 shows the extent to which LIDAR and the triangulation camera can detect individual reflections depending on the position and angles of the (successive) reflections from the water surface, bubbles, or droplets. The results of the individual reflections can vary considerably. Using the time series of the scans and the images captured by the camera, we can compare the mean values of the surface height and the standard deviation of the water level from the mean.

Although the LIDAR and triangulation methods generally perform similarly, they exhibit different behavior when measuring surface profiles of instantaneous water height. LIDAR rejects more measurements for less turbulent surfaces than the laser triangulation method, which typically rejects more measurements for highly turbulent surfaces. In [11], LIDAR proved to be a better localized method than laser triangulation. Measurements with synchronized high-speed camera and LIDAR provide a better understanding of how flying droplets, foam, and bubbles trapped beneath the surface affect the performance of the LIDAR and laser triangulation methods. The study [11] used a novel post-processing algorithm for high-speed imaging that applies epipolar lines to more efficiently distinguish between primary reflections (used for distance measurements at the water surface) and secondary reflections, which introduce noise into the measurements and consequently increase measurement uncertainty. The authors used the same experimental setup as shown in Figure 4. They found that the effects of the imaging parameters (height and width of the filter core and width of the area surrounding the epipolar line) are at least an order of magnitude smaller than the amplitude of the water surface fluctuation. For a review of the literature on LIDAR and triangulation measurements, see Table 2.

### 2.6. Photogrammetric Methods

Unlike triangulation methods, the concept of camera-based photogrammetry is to detect surface topography from the entire image texture rather than from isolated points of incident and reflected laser light sources. The concept of using camera images for the 3D reconstruction of geometric features of solid surfaces and flows is well established in both laboratory and natural environments [69]. Authors of [70] used structure-from-motion photogrammetry for the segmentation, shape and volume determination of large wood assemblages in river systems. Photogrammetry is also an important method for capturing topography and texture in augmented and virtual reality, for example, in the 3D structural modelling of buildings [71] and underground mines [72]. Although photogrammetric measurement methodology was initially developed for measuring the geometry of solids, similar algorithms can be used to reconstruct fluid flow on the free surface. The success of free-surface fluid flow reconstruction depends on many factors, including the use of multiple high-resolution cameras with partially overlapping fields of view, good image sharpness (negligible motion blur and good depth field is required), and the sufficient presence of trackable flow features such as seed particles and illumination points [73]. Water surface photogrammetry is often performed in conjunction with other types of measurements, either as a reference water level measurement or as the main measurement method, e.g., laser triangulation [11].

Free-surface streams where topography reconstruction is less challenging include streams with narrow open channels and spillways where water elevation does not vary greatly across the direction of flow. Flows that can be accurately captured in two dimensions by a single camera do not necessarily require photogrammetric analysis methods but can often be adequately analyzed by simpler optical methods such as edge detection, thresholding for grayscale, and other well-established image transformation methods. Recent reviews of optical methods for measuring the topography of free-surface flows were published in [74,75]. Bung [17] used a high-speed camera to study the air–water surface topography of a stepped spillway model, where the spillway walls were transparent to reveal the flow cross-section. Ferreira et al. [73] developed an automated algorithm to extract the topography of the free surface of a straight channel with an open channel using structure-from-motion/multi-view stereo (SfM-MVS) photogrammetry with seeding particles (floating markers) as visible flow features and three spatially arranged cameras. A similar case with wave channels was studied by Fleming et al. [76], who also used a three-camera setup and particle seeding. The authors simultaneously measured the mean surface height of the fluid and the wave velocity using digital image correlation (DIC). As noted in [75], optical methods based on particle seeding and/or submerged 2D calibration targets with patterns perform best when vertical amplitude of the fluctuations at the fluid surface was small compared to the planar dimension of the fluid mass, and there are few or no air pockets.

Studies [73,76] were able to achieve a good measurement accuracy of 1–2 mm, but the robustness and application of underlying measurement methods is limited to cases where floating particle seeding can be effectively implemented—typically irrotational flows such as low-steepness non-breaking waves flows with low-amplitude surface-level fluctuations and no inflow sources. Hydraulic structure design more often than not involves highly turbulent and aerated flows where seeding particles cannot be implemented and thus requires the development of methods that rely on naturally occurring visual features such as bubbles and droplets. A recent study by Bung et al. [77] employed an RGB-D camera for monitoring and characterizing a highly aerated hydraulic jump using two different methods to calibrate depth estimations (both leading to similar results). The method was able to measure 3D surfaces with high temporal and spatial resolution, although the result evaluation of measurement accuracy was somewhat limited as it was only performed using two-dimensional flow edge position data at channel walls.

However, when the fluid flow is highly turbulent and exhibits similarly large flow fluctuations in all three dimensions, such as in a T-branching flow [11,61], highly complex flow structures with trapped air bubbles and random splashes are formed. The use of external visual aids other than static markers is also limited because submerged calibration targets are too distorted or even obstructed by complex two-phase structures, and buoyant particle markers are washed away from the intended measurement area by downwelling and upwelling currents [76]. In this case, only naturally occurring features on or near the liquid surface (e.g., rims, bubbles, droplets, and light reflections) can be reliably used to detect the free surface.

Few publications have addressed the reconstruction of the topography of such complex flows from image data. As mentioned above, Pavlovčič et al. [11] performed a triangulation of images of the surface structure of flows marked by a scanning laser from a LIDAR system and obtained good agreement with LIDAR topography measurements. Jašarević et al. [78] attempted a true photogrammetric free-surface shape reconstruction of a 90° confluent flow using an array of two high-speed cameras with partially overlapping fields of view. Although the authors succeeded in obtaining a 3D surface model of the standing wave, the model accuracy was poor due to insufficient depth perception. The reason for the rather poor performance of the photogrammetry method in the case of [78] is due to the specular reflection of the water and the small number of cameras used. Namely, since the cameras view the 3D surface only from slightly different angles due to the specular reflections, they can detect significantly different features. To some extent, this can be remedied by illuminating the surface as uniformly as possible.

To more accurately measure the topography of complex free-surface flows using photogrammetry alone (i.e., without support from other methods such as laser triangulation), future research will require an array of multiple fully synchronized, high-resolution cameras supported by algorithms similar to those already used to reconstruct the topography of solid bodies (Figure 5). Unlike laser scanning methods that obtain topography directly from the measured distance between the light source and the fluid-free surface [32], photogrammetry involves several steps before the surface model can be created. Several reconstruction methods have been developed, ranging from methods using a single camera, such as shape from shading, texture, or focus, and methods using a spatial array of multiple cameras [63]. Due to the lack of permanent shape and distinct visual features, flows with free surfaces typically require reconstruction methods that use multiple cameras, especially when the flow is turbulent and highly aerated.

There are two possible approaches, namely, the stereo and silhouette reconstruction algorithms [63]. The stereo reconstruction algorithm is based on the detection of visible topographic features in the flow texture, such as bubbles, droplets, or artificially generated laser light reflections, which are introduced to ensure the correct and robust triangulation of the illuminated regions [11]. The silhouette reconstruction algorithm, on the other hand, works on the principle of intersecting many different camera views, obtaining only the outer edges of the observed flow, and then generating a 3D surface, also known as a silhouette. When scanning solid objects, the silhouette algorithm is often more robust than the stereo algorithm, but at the cost of less detailed topography without color and texture data [63]. However, in the case of free-surface flows, the use of the silhouette algorithm is limited due to the limited number of camera positions, especially in the case of crossing flows where the walls of the flow channels obstruct the flow and only allow the computation of very coarse silhouettes without accurate local height data. For this reason, the stereo algorithm is better suited for reconstructing the surface topography of the flow and requires the following steps to be performed [63]:Identifying and marking unique image features such as edges, bubbles, droplets and light reflections. Markers visible in all images can be used when naturally occurring features are insufficient or unreliable.The matching of labeled features between images from different cameras and time points. High-quality matches represent a linkage of features between many different images.Camera Alignment. Using the positions of the image features from the previous step, camera positions can now be calculated.Generating a 3D point cloud by projecting the matched image features into a three-dimensional space using the camera positions determined in the previous step. The basis for a point cloud are the nodes resulting from the projections of points that intersect on the same features in multiple images.Mesh generation from the point cloud, e.g., using radial basis functions, interpolation of distributed data and implementation of mesh elements (e.g., tetrahedral or other types).Application of texture and color information to 3D meshes from 2D images.

Research is currently underway at the Faculty of Civil Engineering and Geodesy in Ljubljana to reconstruct the topography of free-surface flows from turbulent confluences using the algorithm presented above (see Acknowledgements for funding details). Some promising (though not yet published) results have recently been obtained using an array of 10 synchronized monochrome cameras with a high degree of overlap. As shown in Figure 5, images of relatively modest resolution (1.3 megapixels) were successfully processed in RealityCapture software to create a 3D mesh of the free surface of the confluence. Depending on camera field-of-view coverage, there were both high- and low-resolution regions of the reconstructed free flow surface.

The main advantage of texture-based photogrammetry compared to LIDAR scanning is that the latter method does not require moving parts and can measure topography, with the possibility of reapplying textures. On the other hand, the LIDAR technology only provides surface distance and remission data and requires a mechanical or solid-state system to manipulate the beam position along the third scan dimension (not a problem if the flow profile is scanned in time along a single fixed laser scan line). This limits the 3D scanning capability to flows that have a steady-state character (e.g., standing waves) because multiple surface elevation profiles must be measured sequentially [32] and the result is a smoothed (time-averaged) free-surface shape. Nevertheless, the LIDAR technology is well established and quite robust in measuring complex aerated flows, while photogrammetric 3D water surface measurement methods are still in their infancy and require further improvement and validation [78].

## 3. Conclusions and Perspective

This article reviews conventional and modern optical methods for measuring the topography of free water. Although reliable and easy to implement, conventional point-based measurement methods such as pressure gauges, wave probes, and point gauges encounter serious limitations when measuring the free surface of turbulent and aerated flows. Their application is particularly limited when the flow is both non-steady and three-dimensional (e.g., at a confluence).

In contrast, optical methods previously developed for various terrestrial applications have demonstrated the ability to measure free surfaces with high spatial and temporal resolution. Research is now focused on LIDAR -based laser ranging and laser triangulation, which have been shown to be able to measure turbulent and aerated 3D flows in a time-dependent manner with high speed and resolution. In addition to LIDAR, photogrammetric methods based on multiple camera arrays with partially overlapping fields of view have recently provided some promising results when combined with 3D shape reconstruction algorithms. We strongly believe that all three of the above optical methods have great value for future research.

Nevertheless, we must point out differences in the operation and performance of the various optical methods and compare them to conventional methods. The lidar beam is reflected on the aerated water surface in a series of specular reflections, some of which eventually reach the lidar receiver. A series of specular reflections, together with an abundance of droplets and bubbles on the surface and an often much larger beam area compared to droplets and bubbles, can be called quasi-diffuse reflection. Interestingly, after a series of reflections, the received beam may return to the receiving optics from a slightly different location than the emitted beam that incidents the water surface.

It is also worth noting that novel measurement methods cannot be easily calibrated or validated. We propose to select a standard test case and use it for comparison between different optical measurement methods and research laboratories. For this purpose, we propose a 90° confluence of two flows with high Froude and Reynolds numbers. The reason for the selection is the relative simplicity of the measurement station, the availability of measurement results, the possibility to test scenarios for main and side flows and providing for a variety of quasi-standing waves, including those with a concave surface. The test case can also be used to test the performance of various CFD algorithms and software.

## Figures and Tables

**Figure 1 sensors-23-01842-f001:**
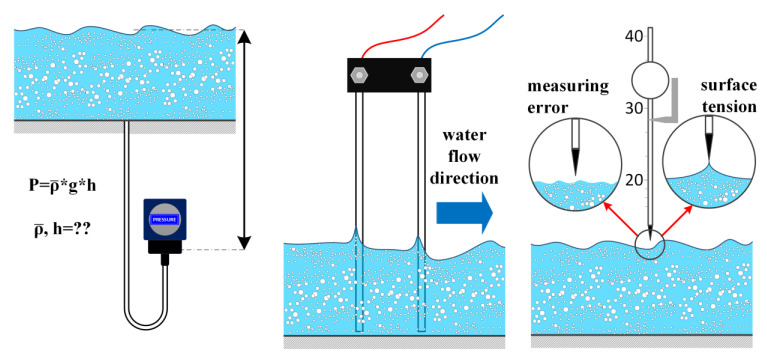
Principle of operation of manometers (**left**), wave probes (**middle**) and point gauge (**right**), with common issues found.

**Figure 2 sensors-23-01842-f002:**
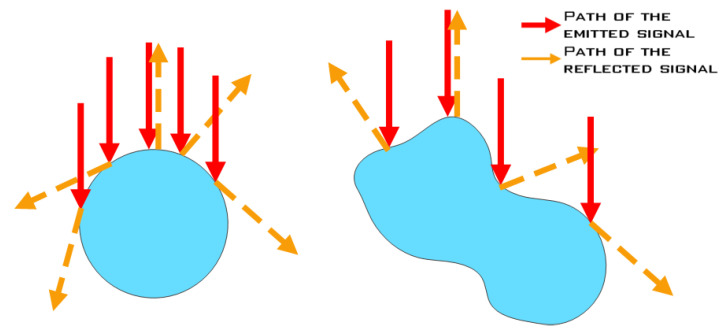
Specular reflections from individual bubbles of different shapes.

**Figure 3 sensors-23-01842-f003:**
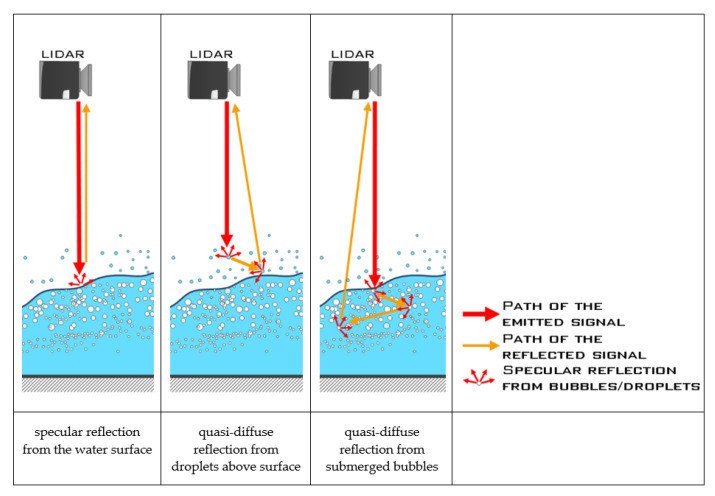
Different types of laser beam reflections during LIDAR measurements.

**Figure 4 sensors-23-01842-f004:**
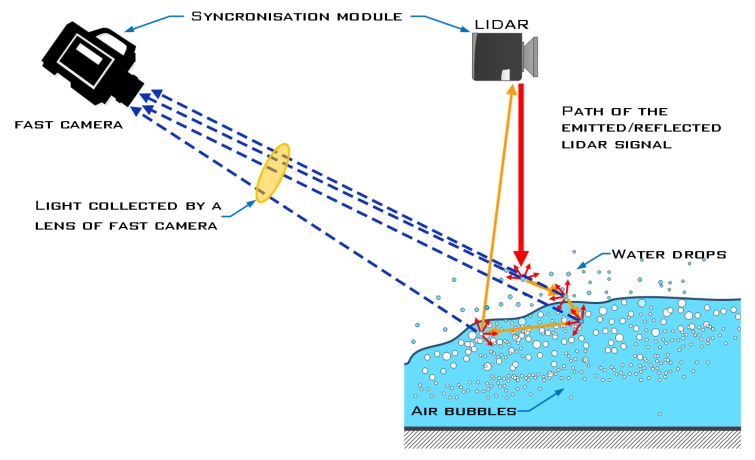
Specular reflections of the laser scanner beam on an aerated water surface [11].

**Figure 5 sensors-23-01842-f005:**
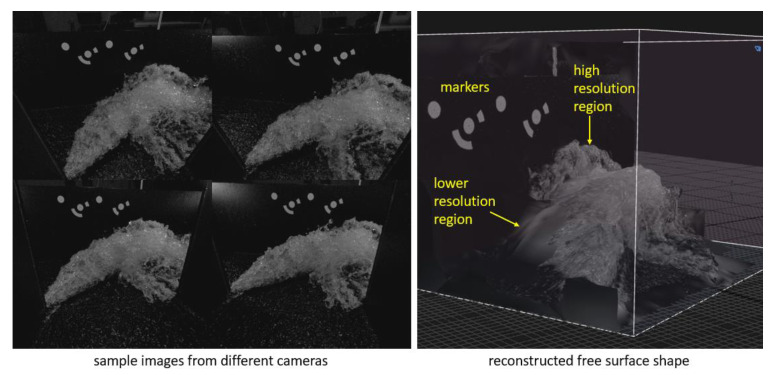
3D free-surface shape reconstruction from partly overlapping 2D images.

**Table 1 sensors-23-01842-t001:** General classification of free-water-surface measurement methods.

Method	Contact	Non-Contact	Point Measurement	2D/3D Measurement	Low Sampling Rate	High Sampling Rate
Manometers	x		x		x	
Wave probes	x		x			x
Point gauges	x		x		x	
Electromagnetic sensors	x		x			x
Ultrasonic sensors		x	x			x
Laser ranging and triangulation		x	x	x		x
High-speed imaging		x		x		x

**Table 2 sensors-23-01842-t002:** Overview of studies using LIDAR and triangulation for free-surface measurements.

Authors	Ref. No.	Device Type	Lab./Field Meas.	Application	Scanning Frequency [Hz]	Angular Resolution [°]	Time-Averaged Profile	Fluctuation	Particulate Matter Added	Foamed/Aerated/Non-Aerated Flow
Blenkinsopp et al.	[49]	LIDAR	Field	Breaking waves	37.5	0.5	no	no	no	foamed
Allis et al.	[67]	LIDAR	Lab.	Wave propagation	37.5	0.5	no	no	yes	non-aerated
Blenkinsopp et al.	[50]	LIDAR	Lab.	Wave propagation	37.5	0.5	no	no	yes	non-aerated
Streicher	[68]	LIDAR	Lab.	Wave propagation	37.5	0.5	no	no	yes	non-aerated
Martins et al.	[52]	LIDAR	Field	Breaking waves	25	0.17	no	no	no	foamed
Martins et al.	[53]	LIDAR	Field	Undular bore	25	0.17	no	no	no	foamed
Martins et al.	[54]	LIDAR	Field	Breaking waves	25	0.17	no	no	no	foamed
Rak et al.	[8]	LIDAR	Lab.	Supercritical junction flow (standing waves)	25–268	0.1–0.25	yes	no	no	aerated
Rak et al.	[60]	LIDAR	Lab.	Supercritical junction flow (standing waves)	268	0.1	yes	yes	no	aerated
Montano et al.	[55]	LIDAR	Lab.	Hydraulic jump	35	0.25	yes	yes	no	aerated
Kramer et al.	[19]	LIDAR	Lab.	Stepped spillway	35	0.25	yes	yes	no	aerated
Rak et al.	[32]	LIDAR	Lab.	Supercritical junction flow (standing waves)	268	0.1	yes	yes	no	aerated
Rak et al.	[61]	LIDAR + high-speed camera (laser triangulation)	Lab.	Supercritical junction flow (standing waves)	268+ video recording with 100 kfps	0.1	yes	yes	no	aerated
Pavlovčič et al.	[11]	LIDAR + high-speed camera (laser triangulation) + epipolar lines	Lab.	Supercritical junction flow (standing waves)	268+ video recording with 100 kfps	0.1	yes	no	no	aerated
Li et al.	[46]	LIDAR	Lab.	Stilling basin	35	0.25	yes	yes	no	aerated
Li et al.	[59]	LIDAR	Field	Creek	35	0.25	yes	no	no	aerated

## Data Availability

No new data was created.

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
