# Peer review of "A Review on Methods for Measurement of Free Water Surface"

_sensors, 2023, doi:10.3390/s23041842_

Round 1
Reviewer 1 Report
The manuscript reviews methods of measurement of free water surface. The conventional methods and optical methods based on LIDAR laser scanning and photogrammetric reconstruction were reviewed. And advantages and disavantages of the methods were assessed. The authors have done a lot of work. In addition, there were still some doubts that need to be clarified:
(1) Line 33-34: Here are some different fonts. It is recommended to follow the requirements.
(2) Line 118: I suggest adding some content to Table 1, such as accuracy, validity.
(3) Line 214-225: The limitations of ultrasonic sensors need to be described more.
(4) Line 389: Paragraph format is inconsistent.
(5) Line 418: Table 2 is not beautiful.
(6) Line 528: Paragraph format is inconsistent.
Author Response
Please consider attached Word file with our response.

Reviewer 2 Report
This is a review article highlighting an important piece of work in capturing and measuring the complex behavior of turbulent flow on free surface. The overall structure of the paper is well thought through, with some good table to summarize the current state of art used in measuring free water surface.
The reviewer hopes that the authors can consider the following few points, before being considered for publications.
1. The analysis of the reviews need to go deeper. Besides describing the methodology, the authors need to present results of the methodology described, so as to highlight the accuracy/deficiency of the current state of art, and what exactly needs to be addressed. This is currently lacking in this paper, and is unfortunately very important and critical.
2. A quick literature search shows that there are so many recent papers that is not cited in this review article. Given that this is a review article, most recent work should be also looked at for comparison. For example:
a. For LIDAR based papers: 'turbulent free-surface monitoring with an RGB-D sensor: the hydraulic jump case'; 'laser ranging measurements of turbulent water surfaces'
b. For high speed camera on water droplets: 'Effects of liquid viscosity on bubble formation characteristics in a typical membrane bioreactor'
c. For electromagnetic: 'https://doi.org/10.1007/s11356-019-07141-3 '
To name a few.
3. Section 2.6 against show the visual capturing using Photogrammetric methods, but again no quantitative analysis was present. This again should be re-looked to add more substance and robustness to the review article.
Author Response

(The authors gave the same response as above.)

Round 2
Reviewer 2 Report
The authors have addressed the comments adequately, and the paper is ready for publication, pending the editor's decision.